# CART*math*—A Mathematical Model of CAR-T Immunotherapy in Preclinical Studies of Hematological Cancers

**DOI:** 10.3390/cancers13122941

**Published:** 2021-06-11

**Authors:** Luciana R. C. Barros, Emanuelle A. Paixão, Andrea M. P. Valli, Gustavo T. Naozuka, Artur C. Fassoni, Regina C. Almeida

**Affiliations:** 1Center for Translational Research in Oncology, Instituto do Câncer do Estado de São Paulo, Hospital das Clínicas da Faculdade de Medicina ds Universidade de São Paulo, São Paulo 01246-000, Brazil; 2Graduate Program, Laboratório Nacional de Computação Científica, Petrópolis 25651-075, Brazil; earantes@lncc.br (E.A.P.); naozuka@lncc.br (G.T.N.); 3Computer Science Department, Universidade Federal do Espírito Santo, Vitória 29075-910, Brazil; avalli@inf.ufes.br; 4Institute for Mathematics and Computer Science, Universidade Federal de Itajubá, Itajubá 37500-903, Brazil; fassoni@unifei.edu.br; 5Computational Modeling Department, Laboratório Nacional de Computação Científica, Petrópolis 25651-075, Brazil; rcca@lncc.br

**Keywords:** three-population mathematical model, CAR-T lymphocytes, memory CAR-T cells, long-term immunity, tumor-induced immunosuppression

## Abstract

**Simple Summary:**

CAR-T cell immunotherapy uses engineered T lymphocytes to recognize cancer antigens and to directly attack cancer cells and have been successfully used against cancers of hematopoietic origin. New CAR designs involving different and multiple target antigens and co-stimulatory structures are subject to recent developments to circumvent resistance, ultimately requiring intensive preclinical experiments. In this work, we develop a mathematical platform to enable in silico experiments to investigate the interplay between tumor cells, effector, and memory CAR-T cells in immunodeficient mouse models of hematological cancers. The CART*math* software and model serve as tools for analyzing different therapeutic scenarios, such as dosing protocols and a blockade of immune checkpoints, contributing to delineating and reducing preclinical experiments, with the aim of improving therapy efficiency.

**Abstract:**

Immunotherapy has gained great momentum with chimeric antigen receptor T cell (CAR-T) therapy, in which patient’s T lymphocytes are genetically manipulated to recognize tumor-specific antigens, increasing tumor elimination efficiency. In recent years, CAR-T cell immunotherapy for hematological malignancies achieved a great response rate in patients and is a very promising therapy for several other malignancies. Each new CAR design requires a preclinical proof-of-concept experiment using immunodeficient mouse models. The absence of a functional immune system in these mice makes them simple and suitable for use as mathematical models. In this work, we develop a three-population mathematical model to describe tumor response to CAR-T cell immunotherapy in immunodeficient mouse models, encompassing interactions between a non-solid tumor and CAR-T cells (effector and long-term memory). We account for several phenomena, such as tumor-induced immunosuppression, memory pool formation, and conversion of memory into effector CAR-T cells in the presence of new tumor cells. Individual donor and tumor specificities are considered uncertainties in the model parameters. Our model is able to reproduce several CAR-T cell immunotherapy scenarios, with different CAR receptors and tumor targets reported in the literature. We found that therapy effectiveness mostly depends on specific parameters such as the differentiation of effector to memory CAR-T cells, CAR-T cytotoxic capacity, tumor growth rate, and tumor-induced immunosuppression. In summary, our model can contribute to reducing and optimizing the number of in vivo experiments with in silico tests to select specific scenarios that could be tested in experimental research. Such an in silico laboratory is an easy-to-run open-source simulator, built on a Shiny R-based platform called CART*math*. It contains the results of this manuscript as examples and documentation. The developed model together with the CART*math* platform have potential use in assessing different CAR-T cell immunotherapy protocols and its associated efficacy, becoming an accessory for in silico trials.

## 1. Introduction

Adoptive cell therapies have been considered a major advance in the fight against several cancers, especially those associated with the hematopoietic system [1]. Chimeric antigen receptor T (CAR-T) cell immunotherapy is an adoptive cellular therapy in which T lymphocytes are taken from the patient’s blood, genetically modified to recognize specific antigens expressed by the tumor, submitted to in vitro expansion, and reinjected into the patient. Insertion of the CAR gene into T lymphocytes bestows their ability to recognize tumor antigen and to directly attack tumor cells regardless of human leukocyte antigen presentation [2]. Current and future advances in the engineering of CAR and new immune checkpoint inhibitor drugs offer promising perspectives in the treatment of cancer. Due to its verified success in eliminating or relieving endurable types of lymphomas and leukemia, in 2017, the Food and Drug Administration (FDA) approved the commercialization of two therapies with CAR-T cells for the treatment of CD19${}^{+}$ B cell malignancies [3,4]. Other target proteins have been studied recently, such as CD123, which is expressed in many hematological malignancies, including acute myeloid leukemia (AML), Hodgkin’s lymphoma (HL), and acute lymphoblastic leukemia (ALL), among others, which makes it a potential antineoplastic target [5]. Several new targets are under investigation and should be tested on mouse models before clinical trials. A rational experimental design could be achieved with in silico simulations that could point out the most promising scenarios [6]. More importantly, they can be used to reduce in vitro/in vivo experiments as a substitute for laboratory xenograft mouse studies. This requires, at a first step, the development of mathematical models to accurately describe experimental data already present in the literature.

Recent in vitro/in vivo experimental studies investigated the relationship between immunotherapy with CAR-T cells and the development of immunological memory [7,8,9,10]. Using an immunodeficient mouse model, Ruella et al. [7] showed that CAR-T 123 therapy can eliminate HL and can provide long-term immunity against a challenge of the same tumor. Immune checkpoint blockade (ICB) associated with CAR-T cell therapy is also under investigation in mouse models where CAR-T cell therapy fails. With the use of immunodeficient mouse models, Ninomiya et al. [11] showed that tumor-expressing indoleamine 2,3-dioxygenase (IDO), an intracellular enzyme that has an inhibitory effect on T cells, can be better controlled by combining CAR-T cell therapy with 1-methyl-tryptophan (1-MT), an IDO inhibitor. By the end of 2016, four different ICB drugs were also approved for the treatment of lymphoma and melanoma, among other cancers. Although the success of CAR-T cell therapy against hematologic cancers is promising, the mechanisms associated with failures have been reported and are the subject of recent investigations [12]. Notably, many challenges remain to be addressed to improve response rates such as minimum effective CAR-T cell dose, selection of CAR-T subtypes, adverse effects management, combination of therapies, formation and maintenance of immunological memory, suppressive microenvironment, and patient specificity, to mention a few [13]. In this context, mathematical models may contribute to understanding the factors involved in malignant transformation, invasion, and metastasis as well as to examining the responses to therapies [8,14,15,16,17,18,19], to confronting hypotheses, and to testing different settings [20,21,22].

Simplified mathematical models can be used to investigate some of those issues and have several advantages, such as reduced simulation time, which allows for testing several experiments in a relatively short period, and a gain in interpretability, that is, understanding all terms of the model and their impact on the results. Several mathematical models in the literature use predator–prey dynamics to explore CAR-T cells’ kinetics [23,24]. However, most of these models do not consider the complex dynamics of effector CAR-T cell differentiation into memory CAR-T cells and then back to effector cells after antigen recognition. In this work, we focus on the development of a simple mathematical model using a system of three ordinary differential equations (ODEs) to describe CAR-T and tumor cell populations dynamics in immunodeficient mouse models. Specifically, our model encompasses interactions between tumor cells, effector, and long-term memory CAR-T cells. The assessment of donor and tumor specificities is considered uncertainties in the parameter values. We calibrated some of the model parameters with in vitro and in vivo data presented in [7] for CAR-T 123 immunotherapy against the HDLM-2 cell line. Considering that the model parameters are highly uncertain, we built a virtual population that reflects the variability of the control data, allowing us to explore the factors that impact therapy outcomes. We also used the model to retrieve a different CAR-T cell immunotherapy scenario, using data from [11] for CAR-T 19 immunotherapy on RAJI tumors. We remark that changing model parameters appropriately makes the model suitable for application in scenarios with different CAR and cell types. Of note, the developed model requires the inclusion of other mechanisms in future studies to enable its extension for human patients. Our model is implemented as an open-source program, called CART*math*, available at github.com/tmglncc/CARTmath (accessed on 4 May 2021). Using the CART*math* virtual laboratory, a researcher without a mathematical background can test the proposed model, can reproduce our results, and can perform new tests. We use CART*math* to complement the present study by investigating, in silico, the occurrence of different therapy outcomes depending on the relationship between the tumor burden and CAR-T cell number, the intensity of immunosuppression mechanisms in the tumor microenvironment, and intrinsic individual specificities. Our model simulations provide insights into the role of these critical mechanisms on the effectiveness of CAR-T cell immunotherapy, showing that CART*math* can be used for assessing different CAR-T cell immunotherapy protocols and its associated efficacy, complementing and potentially avoiding further in vivo experiments.

## 2. Mathematical Model

### 2.1. Model Development

Within the context of CAR-T cell immunotherapy in immunodeficient mice, the therapy response depends mainly on factors such as the capacity of CAR-T cells to kill tumor cells, the formation of long-term immunological memory, and the immunosuppressive effects of the tumor microenvironment. To address these phenomena, we developed a mathematical model based on ODEs, encompassing three cell populations: the tumor cells (*T*), effector CAR-T cells ($C_{T}$), and memory CAR-T cells ($C_{M}$). Figure 1 shows a schematic description of the mechanisms considered in the model, while the biological meaning of the model parameters is summarized in Table 1.

The dynamics of the effector (activated) CAR-T cells is described by the following:(1)$$\frac{dC_{T}}{dt}=\phi C_{T}-\rho C_{T}+\theta TC_{M}-\alpha TC_{T}\;.$$

The first right-hand side term of (Equation 1) specifies that effector CAR-T cells undergo expansion due to proliferation at a rate of ϕ. This population is reduced at a rate ρ, which includes the natural death of effector CAR-T cells and its differentiation into long-term memory CAR-T cells [25,26] according to the linear progression model described in [8,27]. The term $\theta TC_{M}$ describes the activation of memory CAR-T cells into the effector state due to contact with tumor cells. Indeed, it is well-known that memory CAR-T cells may provide long-lasting protection to the specific tumor/antigen [28,29]. At any future time in which memory CAR-T cells come into contact with the same tumor cells, they can rapidly be converted into effector CAR-T cells, readily activated to prevent tumor progression when enough memory cells are present in the system. It is also known that memory CAR-T cells have a lower activation threshold, which eases the secondary response to a future tumor recurrence [30]. Finally, the term $\alpha TC_{T}$ models the combined effects of stimulatory and inhibitory signals on effector CAR-T cells modulated by the tumor. A negative value of α indicates that effector CAR-T cells undergo expansion stimulated by the tumor burden. On the other hand, a positive value indicates effector CAR-T cell inhibition due to immunosuppressive mechanisms such as the PD1/PD-L1 immune checkpoint [31]. Many other immune checkpoint molecules have already been described, such as IDO, LAG3, and VISTA, with high potential to be used as target therapy [32,33]. IDO is an intracellular enzyme that has an inhibitory activity on T cells and is overexpressed in several human cancers [34,35]. In this work, we consider that inhibitory signals prevail, resulting in positive values for α, and we specifically consider the effects of IDO inhibition later on. Moreover, we assume that a given dose of effector CAR-T cells is introduced into the system as an adoptive therapy.

The dynamics of the immunological memory CAR-T cells, a key of the adaptive immune system [8,36], is modeled by the following:(2)$$\frac{dC_{M}}{dt}=\epsilon C_{T}-\theta TC_{M}-\mu C_{M}\;.$$

Equation (Equation 2) assumes that memory CAR-T cells are formed exclusively from the differentiation of effector CAR-T cells at a rate of ϵ. When in future contact with the same antigen-bearing cancer cells, memory CAR-T cells immediately return to the effector CAR-T cell phenotype at a *per capita* rate proportional to the tumor burden ($\theta TC_{M}$). The term $\mu C_{M}$ describes the natural mortality of memory CAR-T cells, with a rate of μ and mean lifetime 1/μ.

The response of tumor cells to CAR-T immunotherapy is modeled by the following:(3)$$\frac{dT}{dt}=rT(1-bT)-\gamma C_{T}T\;.$$

In the absence of immunosurveillance, we assume a density-dependent growth of cancer cells due to the limitation of available resources in the tumor microenvironment, characterizing the existence of intraspecific tumor cell competition. Tumor growth is described using a logistic growth with a maximum growth rate of *r* and a carrying capacity of 1/b [37,38]. Finally, we assume that effector CAR-T cells kill tumor cells upon contact at a constant *per capita* rate γ; this anti-tumor cytotoxicity mechanism is modeled by the term $\gamma C_{T}T$ [39,40,41].

All parameters assume positive values. Furthermore, based on reasonable biological assumptions, we impose two additional conditions on the model parameters as follows. First, we note that parameter ρ may be written as ρ=η+ϵ, where η is the natural mortality rate of effector CAR-T cells and ϵ is the rate of memory formation. We assume that ϕ>η, which reflects the premise that the healthy donor CAR-T cells are likely to proliferate in vivo and to differentiate into memory CAR-T cells, instead of naturally vanishing. Such condition may be rewritten as ϕ>ρ−ϵ. Second, we note that, in general, when the CAR-T therapy leads to complete remission, the tumor is eliminated in a few days and the populations of effector and memory CAR-T cells decrease over time. Additionally, while effector CAR-T cells have a short life span and are not detected on peripheral blood analyses after tumor elimination [7,11], memory CAR-T cells can survive for years [42], providing long-term protection against the target antigen presented by the tumor. This biological behavior is obtained by imposing the restriction ϕ<ρ, which ensures that effector CAR-T cells decay to zero in the absence of tumor cells. Table 1 summarizes the two restrictions imposed on the values of the parameters.

### 2.2. In Vitro and In Vivo Data and Model Inference

Parameter estimation was performed for two different scenarios of immunotherapy on immunodeficient mice: CAR-T 123 cells with the HDLM-2 tumor cell line and CAR-T 19 cells on the RAJI tumor cell line, described in [7] and [11], respectively. The latter was subdivided into the following scenarios: wild-type RAJI (RAJI-control) treated with CAR-T 19 immunotherapy, and RAJI expressing IDO enzyme (RAJI-IDO${}^{{}^{+}}$), treated with CAR-T 19 only and with a combined therapy of CAR-T19 together with IDO inhibitor 1-MT. We used in vitro and in vivo data published in [7] and [11] to calibrate the tumor growth rate (*r*) and the CAR-T cytotoxicity activity (γ) for HDLM-2 and RAJI cells. We also calibrated the carrying capacity (1/b) in the HDLM-2 scenario. Due to the lack of experimental data from effector and memory CAR-T cells, all other model parameters were estimated through model simulations. This means that extensive simulations were performed by fixing *r*, *b*, and γ until a (non-unique) set of parameters that can depict the outcomes predicted in the data was found. It is also worth mentioning that tumor burden was experimentally evaluated based on in vivo bioluminescence imaging (BLI) measurements. We considered one BLI unit as one cell. Although we did not find any correspondence in the literature to convert BLI to cell number, BLI is directly correlated with the total number of cells, as shown in [43]. The cytotoxic activity of CAR-T on tumor cells was retrieved from a standard in vitro 4-h chromium-51 release assay [44]. For the RAJI tumor, inference of the CAR-T cell inhibition due to interactions with the tumor (α) was performed based on data from [11]. All of the used data were extracted using the free software G3Data Graph Analyzer [45]. For completeness, these data and details about Bayesian parameter estimation are presented in the Appendix A. The parameter values used in the simulations for each immunotherapy scenario are given in Table 2.

### 2.3. Mathematical Analysis of Model Dynamics

We perform a mathematical analysis of the model long-term dynamics, finding the steady states and characterizing their stability. In order to simplify the calculations, we non-dimensionalize systems (Equation 1)–(Equation 3) by setting $C_{T}=\frac{r}{\gamma }X$, $C_{M}=\frac{r}{\gamma }Y$, $T=\frac{1}{b}Z$, and $t=\frac{1}{r}\tau$, where *X*, *Y*, *Z*, and τ are dimensionless variables. Note that *Z* represents a fraction of the *T* tumor cell population with respect to the carrying capacity. The dimensionless system is given by
(4)$$\begin{array}{cc} \frac{dX}{d\tau } & =-pX+qZY-sZX\;, \end{array}$$
(5)$$\begin{array}{cc} \frac{dY}{d\tau } & =uX-qZY-wY\;, \end{array}$$
(6)$$\begin{array}{cc} \frac{dZ}{d\tau } & =Z(1-Z)-XZ\;, \end{array}$$
where $p=\frac{\rho -\phi }{r}$, $q=\frac{\theta }{br}$, $s=\frac{\alpha }{br}$, $u=\frac{\epsilon }{r}$, and $w=\frac{\mu }{r}$; note that these parameters are positive due to the conditions imposed on the original parameters (see Table 1). Systems (Equation 4)–() have the following steady states. The trivial equilibrium point corresponding to tumor elimination is
$$P_{0}=\left(0,0,0\right)\;.$$

Another equilibrium point, representing the tumor escape, is given by
$$P_{1}=\left(0,0,1\right)\;.$$

Finally, there are also two nontrivial equilibria corresponding to the coexistence between tumor cells, effector, and long-term memory CAR-T cells, given by
$$P_{i}=\left(X_{i},Y_{i},Z_{i}\right)=\left(X_{i},\frac{\left[\vartheta -s\left(1-X_{i}\right)\right]X_{i}}{w},1-X_{i}\right),\;\;i=2,3\;,$$
where ϑ=u−p, with u−p being positive due to the first condition imposed on the original parameters, and where $X_{2}$ and $X_{3}$ are the roots of the second-degree equation
$$aX^{2}+bX+c=0\;,$$
with coefficients
a=qs>0,b=qr−2qs−sw,c=pw−qr+qs+sw.

Assessing the positiveness and stability of the steady states, we found two thresholds (bifurcation points), given by
$$\vartheta_{T}=\left(p+s\right)\frac{w}{q}+s\;\mathrm{and}\;\vartheta_{SN}=\frac{sw}{q}+2\sqrt{p\frac{sw}{q}}\;.$$

These thresholds determine the following regions in the parameter space where the model presents different dynamic behavior (see Appendix A for details):(I)In region $R_{1}=\left\{\left(\vartheta ,s\right);\;0<s\leq \frac{pw}{q},\;\vartheta <\vartheta_{T}\right\}⋃\left\{\left(\vartheta ,s\right);\;s\geq \frac{pw}{q},\;\vartheta <\vartheta_{SN}\right\}$, the nonnegative equilibria are $P_{0}$ (which is a saddle point) and $P_{1}$ (which is locally asymptotically stable);(II)In region $R_{2}=\left\{\left(\vartheta ,s\right);\;s>0,\;\vartheta >\vartheta_{T}\right\}$, there are three nonnegative equilibria, which are $P_{0}$ (saddle point), $P_{1}$ (saddle point), and $P_{3}$ (locally asymptotically stable);(III)In region $R_{3}=\left\{\left(\vartheta ,s\right);\;s>\frac{pw}{q},\;\vartheta_{SN}<\vartheta <\vartheta_{T}\right\}$, there are four nonnegative equilibria, which are $P_{0}$ (saddle point), $P_{1}$ (locally asymptotically stable), $P_{2}$ (saddle point), and $P_{3}$ (locally asymptotically stable).

The division of the ϑ×s plane into regions $R_{1}$, $R_{2}$, and $R_{3}$ is shown in Figure 2a.

In order to cure a patient, the system trajectory must be either in the basin of attraction of the tumor elimination equilibrium $P_{0}$ or in the basin of attraction of a stable coexistence equilibrium, where only a harmless, small amount of tumor cells is present, as described by equilibrium $P_{3}$. Since the point $P_{0}$ is always unstable, only the last option is possible, which can be reached in regions $R_{2}$ and $R_{3}$. While in region $R_{2}$, the tumor escape equilibrium $P_{1}$ is unstable (and all trajectories eventually converge to equilibrium $P_{3}$), in region $R_{3}$, we have bistability between $P_{3}$ and $P_{1}$; in this case, the model outcome (tumor control or escape) depends on the initial conditions. Setting the parameter values to those calibrated for immunotherapy with HDLM-2 tumor cells (Table 2), the model dynamics corresponds to region $R_{3}$ with $P_{2}=\left(1.5205\times 10^{4},2.63\times 10^{2},1.442522\times 10^{6}\right)$ and $P_{3}=\left(1.5205\times 10^{4},1.97519\times 10^{5},1.089\times 10^{3}\right)$ in dimensional units. The basins of attraction corresponding to elimination/control and escape are shown in Figure 2b along with some trajectories of typical model solutions, some leading to the escape equilibrium point ($P_{1}$) and others to the coexistence equilibrium point ($P_{3}$).

### 2.4. In Silico Population and Sensitivity Analysis

Individual and tumor specificities may lead to different therapy outcomes under the same treatment regime. In terms of modeling, they are represented by variations in the model parameter values. Therefore, it is essential to explore therapy responses by considering a variety of plausible physiological parameter sets. To avoid introducing combinations of parameters that characterize spurious, non-physiological individuals, we rely on building a virtual population (VP) that reflects the variability observed in the available data. For this, we use a strategy similar to that described in [46]. Of note, virtual clinical trials are becoming increasingly popular to represent the heterogeneity of patient cohorts in pharmacology models [47,48]. Here, the resulting VP is used to investigate how population heterogeneity impacts overall treatment responses and to identify the most influential parameters for each of them.

To build the VP, we first assume that each model parameter is a random variable following a uniform distribution with a wide plausible range. We take random samples from the parametric space, each one representing a plausible virtual mouse. The set of accepted parameters must satisfy the restrictions ϕ<ρ and ϵ−ρ+ϕ>0, as indicated in Table 1. Moreover, each physiologically plausible set of parameters is included in the VP only if it leads to a predefined characteristic or behavior similar to that of the target distribution. Specifically, here, we build a VP that matches the overall survival of non-treated NSG mice injected with HDLM-2 cells reported in [7]. This means that we simulate the model for a plausible virtual mouse over 300 days, considered here the maximum life time of untreated mice, and it is accepted as a member of the VP if its survival is in the range of the actual population from [7]. The procedure proceeds until obtaining a VP with 5000 virtual mice (VM), with mean and medium overall survival statistically similar to the actual population.

The VP is then submitted to CAR-T therapy and the overall treatment response is evaluated over 300 days. The therapy outcomes are classified into complete response (CR), when the number of tumor cells is less than or equal to the detection threshold (assumed equal to $8\times 10^{5}$ cells, as indicated in [7]), and non-responder (NR), when the number of tumor cells is greater than $1\times 10^{10}$, considered a lethal tumor burden [7]; for completeness, we classify the outcome with a number of tumor cells between $8\times 10^{5}$ and $1\times 10^{10}$ as partial response (PR). Survival curves are made for different CAR-T doses to investigate how the CR rate decreases with dose reduction. Global sensitivity analysis is carried out for each treatment outcome by examining scatter plots and by evaluating the Pearson correlation coefficient between the chosen model variable (tumor or memory CAR-T cells) and all of the model parameters early after therapy. In this way, we identify the parameters that impact tumor burden and the formation of the immunological memory depending on the therapy response the most.

### 2.5. Model Settings and Numerical Solution

Mathematical equations (Equations (Equation 1)–(Equation 3)) were solved numerically using the explicit fourth-order Runge–Kutta method [49]. The simulations represent CAR-T cell therapy in immunodeficient mice previously injected with tumor cells. The initial condition for the tumor population, T(0), corresponds to the number of injected tumor cells, while for CAR-T cells, we assume $C_{T}\left(0\right)=C_{M}\left(0\right)=0$ cell. At the time, when immunotherapy is given, a CAR-T cell dose is attributed to $C_{T}$ and tumor cells have already undergone significant growth. Cell populations were followed up to investigate tumor response and immunological memory formation. In the numerical solution procedure of the model, the size of the zero cell population threshold is defined as $10^{-6}$ cells. Thus, when any cell population reaches cell numbers below $10^{-6}$, it is treated as extinct by assigning a zero value directly to the corresponding variable. A direct consequence of this hypothesis is that there may be a complete elimination of the tumor, although mathematical analysis indicates that the elimination point $P_{0}=\left(0,0,0\right)$ is always a saddle point. Our model framework is implemented in CART*math* [50], whereby most of the results presented in Section 3 may be easily reproduced using predefined datasets as explained in the CART*math* manual.

## 3. Results: In Silico Experiments

CART*math* was used to simulate the scenarios of CAR-T 123 cell immunotherapy on Hodgkin’s Lymphoma (HLDM-2 cell line) and CAR-T 19 cells immunotherapy on ALL-B (RAJI cell line) in immunodeficient mouse models. HDLM-2 cell line has a low growth rate and can be rapidly eliminated upon CAR-T 123 immunotherapy even in second tumor injection (by challenging previously treated mice). On the other hand, RAJI cells have a very fast growth rate and are not eliminated by CAR-T cells. These two preclinical models that represent two very different scenarios are used here to demonstrate the plasticity of the developed mathematical model. Using in silico experiments, we investigate how parameter uncertainties impact CAR-T 123 immunotherapy outcomes. For the RAJI tumor scenario, we also explore the effect of CAR-T 19 cell immunotherapy alone or combined with ICB therapy. Of note, the ICB therapy combination with CAR-T cells is promising in the case of CAR-T cell therapy resistance and is under investigation in biological studies [51,52].

### 3.1. CAR-T 123 Therapy Eliminates HDLM-2 Tumors, Providing Long-Term Protection, While Immunotherapy with CAR-T 19 on RAJI Tumor Slows Down Its Growth

We first simulated the scenario presented in [7], which consists of CAR-T 123 therapy against HDLM-2 cells. Ruella et al. [7] reported that $2\times 10^{6}$ cells of Hodgkin lymphoma (HDLM-2) were injected into immunodeficient NSG mice. Simulation begins with $T\left(0\right)=2\times 10^{6}$ HDLM-2 cells and tumor progresses in time until it reaches about $2\times 10^{7}$ cells at t=42 days (Figure 3a). At this time, immunotherapy with CAR-T 123 cells is performed, so that we set $C_{T}=2\times 10^{6}$ cells at t=42 days. Effector CAR-T cells rapidly eliminate tumor cells in a few days, retrieving the experimental remission results presented in [7]. Our simulation also provides the dynamics of memory CAR-T cells. Figure 3a shows that, as the population of $C_{T}$ cells decreases, phenotypic differentiation occurs, giving rise to memory CAR-T cells $C_{M}$. Our simulation shows that effector CAR-T cell populations remain undetectable until t=250 days, which agrees with the results presented in [7]. Moreover, our model indicates the presence of long-term memory CAR-T cells, which slightly decline in time due to a small mortality rate of μ. Model parameters values used in this simulation are displayed in Table 2.

In an additional experiment, Ruella et al. [7] demonstrated the formation of immune memory by challenging previously treated mice with $1\times 10^{6}$ HDLM-2 cells at t=250 days. The tumor remained undetectable, being eliminated due to re-expansion of the effector CAR-T cells. To investigate the model behavior with this respect, we continued the previous simulation by introducing $1\times 10^{6}$ tumor cells at t=250 days. Figure 3a shows the model outcome for this challenge. The presence of tumor cells drives the conversion of memory into effector CAR-T cells, which are rapidly able to eliminate the new tumor. Afterward, effector CAR-T cells undergo rapid decay while part of the memory CAR-T cells population is recovered. Tumor clearance remains until the end of simulation on day 500. As explained in [7], tumor rejection occurs due to the re-activation of previously undetectable memory CAR-T cells.

Next, we investigated the model behavior in a different scenario with a fast growth tumor cell. The corresponding experiment is described in [11], which uses RAJI tumor and immunotherapy with CAR-T 19 cells. RAJI tumors are much more aggressive than HDLM-2 tumors and express the CD19 antigen. Ninomiya et al. [11] reported that $3\times 10^{6}$ RAJI tumor cells were injected in SCID/Beige mice and that therapy with $1\times 10^{7}$ cells of CAR-T 19 was given on day 7, which did not eliminate the tumor but could partially control its growth. This scenario was simulated with the estimated parameter values displayed in Table 2. Starting with $T\left(0\right)=3\times 10^{6}$ cells, the tumor reached almost $1\times 10^{8}$ cells on day 7, when $C_{T}=1\times 10^{7}$ cells of CAR-T 19 were introduced. Retrieving the results presented in [11], the immunotherapy was able to reduce the tumor growth rate but not eliminate it, and tumor cell population reached $6\times 10^{8}$ cells on day 14, as shown in Figure 3b. The effector CAR-T cells underwent an expansion of about 30% on day 9, from which they decreased until extinction, representing the CAR-T cell time course reported in [11]. In the original experiment and our model simulation, memory CAR-T cells were not generated.

### 3.2. Insights on Immune Checkpoint Inhibitors

Our model includes the term $\alpha TC_{T}$ in Equation (Equation 1), which describes tumor-modulated immunosuppressive mechanisms. A higher α value implies a stronger immunosuppressive mechanism culminating in less CAR-T cells proliferation. To investigate ICB mechanisms and, at the same time, how the model deals with different tumors and CAR-T cells, we selected data from [11] that presents the action of CAR-T 19 cell immunotherapy against CD19${}^{+}$ lymphoma-expressing IDO in mice. We then considered mice bearing RAJI-IDO${}^{{}^{+}}$ cells treated with CAR-T 19 alone (Figure 3c) or combined with 1-MT (Figure 3d), an IDO inhibitor. We estimated α for these scenarios, keeping all of the other parameters fixed with the values shown in Table 2 for the RAJI-control. According to Figure 2C of [11], it should be noted that RAJI and RAJI-IDO${}^{{}^{+}}$ tumor sizes on the day of immunotherapy administration are indistinguishable so that the same tumor proliferation rate was used in both experiments. The smaller α value obtained when 1-MT was used allowed for a greater expansion of effector CAR-T cells after infusion, which in turn provided a stronger control on the tumor growth. Of note, in both cases, the CAR-T 19 dose is not able to eliminate the tumor, which eventually escapes, and there is no formation of memory CAR-T cells. We can also notice the similarity between α values for the RAJI-control + CAR-T 19 and RAJI-IDO${}^{{}^{+}}$+ CAR-T 19 + 1-MT, reflecting the ability of the 1-MT to block the immunosuppressive effect of IDO. Thus, the model could capture the effect of the IDO inhibitor through the α parameter that can modulate the immunosuppression mechanism used by RAJI-IDO${}^{{}^{+}}$ tumors. These simulations show the ability of α in modulating immunosuppressive mechanisms, displaying the potential use of our mathematical model as an adjuvant in silico platform to test ICB.

### 3.3. Insights on Dosing Strategies: Single and Fractionated Doses

The model is used now to investigate how the relationship between the HDLM-2 tumor burden and CAR-T 123 cell dose and injection protocol impact therapy responses. To first assess how the dose interferes with the response to the CAR-T 123 immunotherapy, we performed three different simulations with therapeutic doses of $1.5\times 10^{6}$, $0.5\times 10^{6}$, and $0.2\times 10^{6}$ cells at t=42 days. We used the same scenario described in Figure 3a and the same model parameters shown in Table 2, keeping the initial tumor burden equals to $T\left(0\right)=2\times 10^{6}$ cells. The resulting dynamics are shown in Figure 4a–c. A CAR-T dose of $1.5\times 10^{6}$ cells can perform tumor elimination, although the level of memory CAR-T cells at t=200 days is smaller than that in the case presented in Figure 3a, in which the therapeutic dose is $2\times 10^{6}$ cells at t=42 days. A higher CAR-T cell dose generates a greater immunological memory CAR-T cell pool. On the other hand, by reducing the CAR-T dose to $0.5\times 10^{6}$ cells, the tumor is not completely eliminated. It undergoes an intense decrease but resumes growth on day 150, eventually reaching a state in which it does not grow or shrink significantly on day 500; the tumor is reduced to a very small (but not zero) value, which characterizes a state of a residual disease, as depicted in Figure 4b. In this immunotherapy outcome, both $C_{T}$ and $C_{M}$ cells are nonzero, and therefore, there is the coexistence of the three cell populations. This is a typical configuration of tumor equilibrium, one of three “Es” of immunoediting [53]. Finally, further reducing the CAR-T dose to $0.2\times 10^{6}$ cells, the tumor escapes (Figure 4c); there is a complete and rapid extinction of the effector CAR-T cell population and no formation of memory CAR-T cells. Remarkably, these three possible immunotherapy responses of elimination, residual disease (coexistence), and escape can also be reached by fixing the CAR-T dose and by increasing the tumor burden.

As the tumor burden in the residual disease outcome at t=300 days is always below the detection threshold, assumed equal to $8\times 10^{5}$ cells [7], we classify both elimination and residual disease responses observed above as CR. On the other hand, since all escape results show tumor burden above the lethal disease threshold of $1\times 10^{10}$ cells at 300 days, they all are classified as NR. Figure 5 shows therapy responses over 300 days to a variety of combinations of CAR-T doses and tumor burden. For a tumor burden of approximately $T\left(0\right)=2\times 10^{6}$ cells, for example, CR is reached with CAR-T doses around $2.6\times 10^{5}$ cells or higher; CAR-T doses lower than $2.6\times 10^{5}$ cells lead to NR. The greater the tumor burden, the greater the CAR-T cell dose needed to achieve CR, which is reflected in the reduction of the CR region in the diagram (Figure 5).

The next experiment explores the alternative possibility of a fractionated treatment using CAR-T cells, which is a strategy tested in the clinic aiming to reduce toxicity effects [54]. We selected the same scenario described in Figure 3a with the 1-time infusion of $2\times 10^{6}$ CAR-T cells, which promotes tumor elimination. First, simulations were performed dividing the total dose into four equal fractions of $0.5\times 10^{6}$, infused every seven or fourteen days. Figure 4d,e show that the dosing split does not interfere with the tumor elimination, which occurs in a few days. Of note, a single dose of $0.5\times 10^{6}$ CAR-T cells is not able to eliminate the tumor burden, as shown in Figure 4b. While in a single infusion case, the tumor decreases but resumes growth until reaching residual disease, the used fractionated infusions prevent tumor regrowth. As in Figure 3a, immunological memory is formed, and the peak of memory cells is similar to that of a single total dose infusion, although a certain delay is observed due to the fractionated dose. Such a delay ultimately yields a greater formation of immunological memory on day 200. Specifically, the number of memory CAR-T cells at that time is around 7% and 15% larger for 7 and 14 days rest time between doses, respectively. Alternatively, a simulation was performed for the fractionated immunotherapy described in [54]. In that work, patients with relapsed or refractory CD19${}^{+}$ ALL were treated with three fractionated infusions over three consecutive days with increasing doses (10%, 30%, and 60%). It was shown that such a treatment protocol does not compromise effectiveness while reducing toxicity effects [54]. Figure 4f shows the in silico predictions using this protocol. As in the single time infusion protocol shown in Figure 3a, the tumor is rapidly eliminated, effector CAR-T cells vanish in 100 days, while immunologic memory amounts to $1.5\times 10^{6}$ cells on day 200.

### 3.4. Insights on Parameter Uncertainties Impacting Treatment Outcome

Our VP was built to reflect the variability observed in the experimental data reported in [7]: five non-treated NSG mice survived from 100 up to 207 days after tumor engraftment, with a mean survival of 137 days. To build our VP, we first defined wide and plausible ranges for the model parameters. Each parameter was assumed to be a random variable with uniform distribution in the range limited by ±60% of the reference values indicated in Table 2 for HDLM-2 + CAR-T 123. This range was crucial to obtain our target VP with mean and median survivals around 137 and 128 days, as observed in [7]. The survival curves for both control data from [7] and the VP are depicted in Figure 6a, displaying statistically similar mean and median survival times. We then submitted our VP to CAR-T 123 treatment with different doses, varying from $1.5\times 10^{6}$ to $1.0\times 10^{5}$ CAR-T cells. While 100% overall survival was reached with mice treated with $1.5\times 10^{6}$ CAR-T 123 cells in [7], the VP reached 95% overall survival in 300 days, corresponding to 4754 VM (see Figure 6b). Such a 5% reduction can be explained by the individual variability in the VP. Figure 6b also shows that the overall survival was significantly reduced with the decrease in the immunotherapy dose. The frequency of parameter values in the VP and their distributions for the two different therapy outcomes are shown in the Appendix A.

We now use in silico experiments to investigate how parameter uncertainties impact the CAR-T 123 immunotherapy outcomes. We selected the scenario in which the VP is treated with $1.0\times 10^{6}$ CAR-T cells. In this scenario, 645 VM were non-responders and died, and 4354 VM achieved CR within 300 days. We then evaluated the correlation between the variability in the VM parameters and the immunological memory formation ($C_{M}$) and tumor burden (*T*) at t=55 and t=75 days for each of these therapy outcomes. These analysis times were chosen because they are within the period in which the reduction of effector cells and the expansion of memory cells are expected to occur.

Figure 7 shows the tornado plots for the CR and NR cases with respect to $C_{M}$ obtained at t=55 and t=75 days, i.e., 13 and 33 days after applying CAR-T therapy. Parameter ϵ that modulates the ability of effector CAR-T cells to differentiate into memory CAR-T cells is the most influential in the early formation of immunological memory when therapy is successful. The tumor growth rate *r* plays a negative major role in memory pool formation. The negative effect of tumor inhibition on effector CAR-T cells modulated by α is also remarkable. For the NR cases, the negative effect of both *r* and θ on $C_{M}$ are the most influential, yielding a growing tumor burden that keeps activating memory CAR-T cells into effector CAR-T cells, ultimately precluding the formation of the immunological memory pool. In general, the correlation values at t=75 days of the mentioned most influential parameters decreased when compared to their values at t=55 days, and we may also note changes in the ranking of the importance of the parameters. The sensitivity analysis with respect to *T* is shown in the Appendix A. The most influential parameter is the tumor growth rate *r*, and the role of the cytotoxic coefficient γ in controlling the tumor burden for the CR cases is also remarkable.

## 4. Discussion

The use of CAR-T cell immunotherapies is spreading across hematological cancers and has already turned into products of big pharma companies [55]. On the road, there are new CAR designs, including new antigen targets [6], different CAR affinity [56], and expansion protocols [57]. Mathematical models can be used as accessory tools for new developments [18,19]. Here, we built a three population mathematical model to describe tumor response to CAR-T cell immunotherapy in immunodeficient mouse models (NSG and SCID/beige) based on two published articles from the literature [7,11]. Our model was able to represent different receptors independently of the recognized antigen, such as CAR-T 19BBζ and CAR-T 123, and different tumor targets such as HDLM-2 and RAJI. The HDLM-2 tumor model was used as a low proliferation, less aggressive tumor model, where CAR-T cell immunotherapy can be effective on tumor elimination and on memory CAR-T cells emergence. On the other hand, the RAJI model was chosen for its high proliferation and escape from CAR-T cell immunotherapy. In this scenario, our model was able to capture the effect of the IDO enzyme expression by the RAJI cells as well as the impact of CAR-T cell immunotherapy and their combination with an IDO inhibitor. This fact reflects the potential of our model for describing other different immune checkpoint inhibitor molecules. Indeed, changing model parameters appropriately would make the model suitable for application in different treatment and tumor scenarios. The conversion of CAR-T cells from effector to memory cells and their long-term persistence as memory CAR-T cells were demonstrated by previous experimental work with the RS4;11 B-ALL model using CAR-T 19BBζ [58]. This biological mechanism was proven to be fundamental in our model for obtaining the outcomes of immunotherapy, highlighting the importance of including memory CAR-T cells in mathematical models. In the HDLM-2 + CAR-T 123 scenario, our model was able to represent tumor elimination after immunotherapy even in the case of a new tumor challenge due to memory CAR-T cells’ long-term protection in the HDLM-2 target. However, the formation of a memory pool was not observed in any of the evaluated RAJI scenarios due to the rapid growth dynamics of this tumor.

We performed in silico studies to highlight how the model could be used as an adjuvant platform to contribute to a better understanding of the underlying processes and for experimental research. Investigating the application of different dosing protocols, we showed that fractionated dose appears to be as effective as a single dose and that the rest periods between infusions might favor long-term immunological memory. These results corroborate previous clinical trials using fractionated CAR-T cell doses with similar effectiveness to single-dose and persistence of CAR-T cells on the blood 20 months after therapy [55]. We also found that the CAR-T cell dose determination for a given tumor burden is a critical factor for the success of the immunotherapy. A previous model already considered CAR-T cell proliferation in response to antigen burden [26], but neither memory CAR-T cells nor the effect of tumor inhibition on CAR-T cells were considered. A recent paper considered naïve, effector, and long-term memory T cells in a refractory large B cell lymphoma model [10]. We did not include naïve CAR-T cells because they pass through an in vitro activation protocol and only activated effector CAR-T cells are present in the treatment [58]. Another interesting mathematical model was made upon tisagenlecleucel-treated patient data [25]. This model was adapted from a previous empirical model of an immune response to bacterial/viral infections. They captured CAR-T cell expansion, contraction, and persistence similar to our model, including memory CAR-T cell population [25]. Their model was calibrated on patients’ data, and contrasting with ours, no difference in dose–response was detected. They attributed this result to the CAR-T cell proliferation capacity in vivo. We partially agree, but there is a possibility that the data obtained from humans does not present very different CAR-T cell doses (especially including only tisagenlecleucel clinical trials). Considering mouse model data, where the CAR-T cell dose varies by thousands, we do observe a dose effect, especially on aggressive, highly proliferative tumors. Another mathematical model was recently published concerning mouse models for breast cancer and CAR-T cells anti-Erb2 [59]. Only tumor and CAR-T cells were considered in the model, and the authors also simulated several CAR-T cells doses based on in vitro and in vivo experiments.

Another advantage of our mathematical model is the therapy effectiveness calculation. Overall therapy effectiveness may depend on intrinsic individual specificities, regarded here as heterogeneity in the values of the model parameters. In the studied case, such parameter uncertainties reduced overall survival in 300 days by 5% for the HDLM-2 + CAR-T 123 scenario. The adopted structure of our mathematical model allows for identifying each mechanism more transparently. Donor/tumor-microenvironment specificities were considered uncertainties in the values of the model parameters, which were shown to greatly impact the therapy outcomes. We identified that uncertainties associated with the tumor proliferation, the ability to inhibit the effector CAR-T cells, tumor cell lysis by CAR-T cells, and the differentiation of effector CAR-T cells into memory CAR-T cells are, among all the mechanisms considered in the model, the most influential in immunotherapy response. This opens room for investigating other chimeric antigen T-cell receptors with different target/antigen affinities, and the blockade of immune checkpoints to boost therapy efficacy and safety. In our model, we did not consider CAR affinity for each antigen as an explicit parameter, considering it as a result of tumor lysis by CAR-T cells. Another aspect that we did not take into consideration is the toxicity effect of CAR-T cell immunotherapy (cytokine release syndrome (CRS)) since our model is based on an immunodeficient mouse model that lacks this effect. Our model does not directly apply to human settings since we do not consider other immune system constituents, especially healthy B cells. By expressing the same tumor antigens, healthy B cells are killed by effector CAR-T cells and seem to play an important role in the in vivo expansion of the CAR-T cells. For human data, Hanson et al. [60] developed a mathematical model for CAR-T cell immunotherapy for B-ALL emphasizing cytokines and CRS, also considering CAR-T effector and memory cells. As an acute effect of CAR-T cell immunotherapy, CRS is caused by effector CAR-T cells hours after the treatment. On the other hand, memory CAR-T cells are correlated with a durable response against the tumor in patients [10] and in mice [7]. Another model [24] explored the competition of CAR-T cells and T lymphocytes for the tumor cells, when both populations are present in patients.

There is still a challenge in CAR-T cell immunotherapy and all cellular therapies, which is the exhaustion of implanted cells. CAR-T cells become exhausted by continuous stimulation from tumor cells harboring the cognate antigen. A recent work modeled CAR-T proliferation and exhaustion using in vitro experimental data from glioblastoma [23]. No spatial distribution was considered in our model, since we are dealing with hematological cancer, but this is required in CAR-T therapy for solid tumors. Difficulties related to access and infiltration in tumors, immunosuppressive mechanisms, and choice of target antigens are among the several challenges in developing successful CAR-T therapy against solid tumors. Recent work investigated CAR-T therapy targeting two antigens against glioblastoma [61]. CARs that incorporate multiple target antigens are also the subject of recent research to overcome the mechanism of resistance to CAR-T therapy [13]. Although not completely understood, the incidence of this phenomenon has been linked to antigen escape or lineage switch [62,63], which can be modeled as stochastic events. A recent mathematical model [10] has already pointed out the importance of considering stochastic events to deal with tumor elimination in response to CAR-T cell therapy. A hybrid technique that combines deterministic and stochastic events was proposed, with the latter included only when tumor cells are under a given threshold that ultimately impacts tumor extinction. This strategy reduces the computational burden associated with the higher cost of stochastic models. However, as stochastic events cannot be neglected in many situations, further researches are still needed towards accurate and computationally efficient methodologies.

Finally, striving for the reproducibility of our results and the expansion of the use of mathematical models and in silico experiments by biologists or any researchers unfamiliar with the mathematical approach, our model has been implemented in a Shiny R-based platform called CART*math*, available at github.com/tmglncc/CARTmath (accessed on 4 May 2021). It provides an in silico tool for assessing different issues associated with the CAR-T immunotherapy such as how CAR-T cell dosing can be adjusted according to tumor burden, CAR-T cell infusion protocols, and immunosuppressive mechanisms, among others, without further in vivo experiments. A quick guide to running and building simulations is provided in the software documentation [50]. We plan to keep on working on the software development, including the integration of new tools such as the one that allows for estimating model parameters to ease integrating new scenarios and the analysis with virtual populations. Overall, the developed mathematical model and CART*math* may help to shed light on the structure of the treatment protocol and in better understanding the challenges that remain in the study of CAR-T cell immunotherapy.

## Figures and Tables

**Figure 1 cancers-13-02941-f001:**
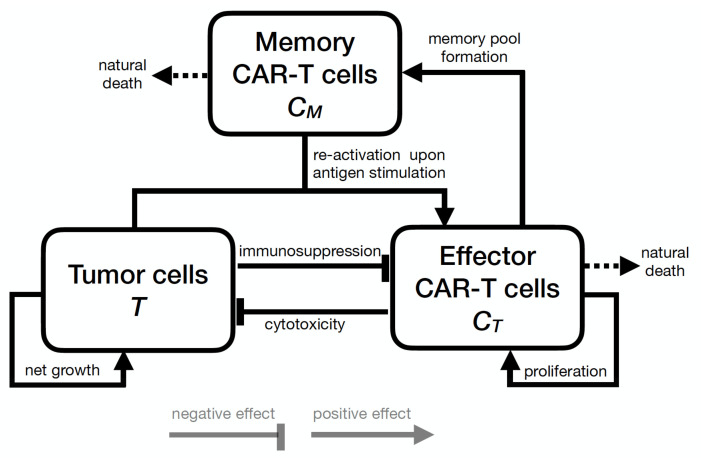
Schematic description of the model structure. Effector CAR-T cells proliferate, have a cytotoxic effect on tumor cells, differentiate into memory CAR-T cells, and die naturally or are impaired by tumor-induced immunosuppressive mechanisms. The long-term memory CAR-T cells also die naturally and are readily responsive to the tumor-associated antigen, and when they interact with tumor cells, they differentiate back into effector CAR-T cells, producing a rapid immune response against the tumor. Tumor cells grow subject to available resources in the microenvironment and are killed by effector CAR-T cells.

**Figure 2 cancers-13-02941-f002:**
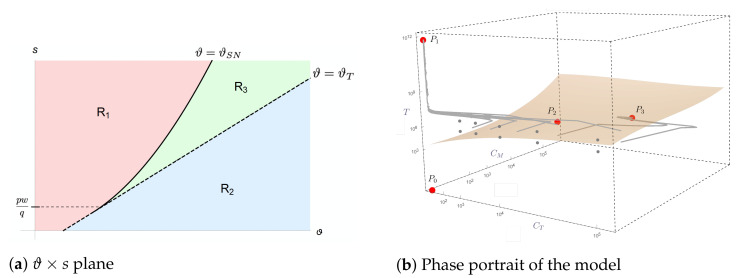
The CAR-T therapy ODE model presents different dynamical behaviors in each of the three regions *R*_1_, *R*_2_, and *R*_3_, indicated in the ϑ × *s* plane (**a**). In the HDLM-2 + CAR-T 123 scenario, the parameter values correspond to region *R*_3_, and the phase portrait in this case together with typical model trajectories are shown in (**b**). The equilibrium points are indicated by red dots. The yellowish surface represents the separatrix between the basins of attraction of *P*_1_ (escape) and *P*_3_ (stable coexistence). The saddle points are indicated by *P*_0_ and *P*_2_.

**Figure 3 cancers-13-02941-f003:**
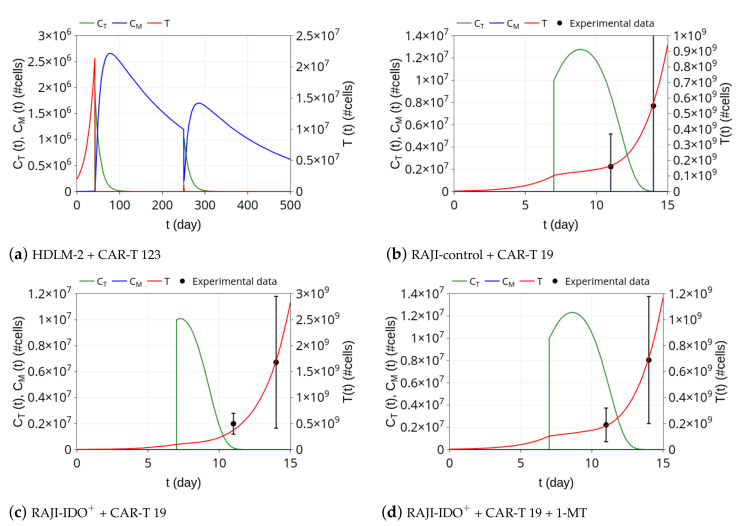
Dynamics of tumor cells *T* (red), effector *C_T_* (green), and memory *C_M_* (blue) CAR-T cell populations (#cells stands for number of cells). (**a**) The immunotherapy with CAR-T 123 on HDLM-2 and challenge were performed at *t* = 42 and *t* = 250 days, respectively. After effector CAR-T cell injection, tumor cells were rapidly eliminated and a decay of effector CAR-T cells is observed, which were partially converted into memory CAR-T cells. The tumor remained undetectable until day 250 when the challenge was carried out. Upon contact with new tumor cells, memory CAR-T cells were converted into effector CAR-T cells, which rapidly eliminated the tumor. Afterward, immunological memory was partially recovered. (**b**) Immunotherapy with CAR-T 19 on RAJI-control was performed on day 7. There was an expansion of effector CAR-T cells, which can reduce tumor growth rate but did not eliminate the tumor. Effector CAR-T cells were extinct at the end of the simulation. There was no memory formation. (**c**) CAR-T 19 immunotherapy on RAJI-IDO^+^ cells. On day 7, 1 × 10^7^ CAR-T 19 cells were introduced and were rapidly eliminated; (**d**) CAR-T 19 immunotherapy with IDO inhibitor (1-MT) shows a restoration of CAR-T cell dynamics, demonstrating the impact of IDO. The parameter *α* was estimated for these two cases and was responsible for capturing the effect of IDO inhibition due to 1-MT. Its value decreased for the RAJI-IDO^+^ + CAR-T 19 + 1-MT case, being small enough to promote a higher expansion of the effector CAR-T cells and ultimately leading to more effective control on the tumor growth. However, both therapies were not able either to eliminate the tumor or build memory cells. Dots and standard deviation correspond to experimental data from [11].

**Figure 4 cancers-13-02941-f004:**
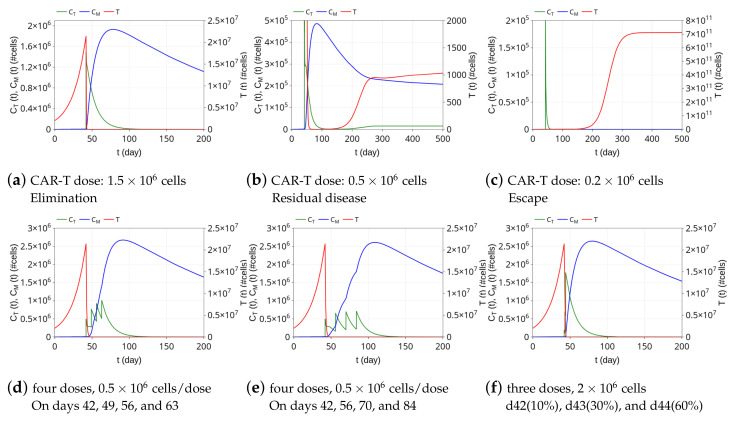
In silico predictions of the immunotherapy response to different CAR-T cell doses and protocols, beginning on day 42. Initial HDLM-2 tumor burden amounts to 2 × 10^6^ cells. Top row: (**a**) with 1.5 × 10^6^ CAR-T cell dose, tumor elimination occurs around day 55 and approximately 7 × 10^5^ memory CAR-T cells remain at *t* = 200 days; (**b**) one third of the previous CAR-T cell dose (0.5 × 10^6^ cells) induces a strong decline in the tumor burden, although tumor rapidly resumes growth. After day 250, the three cell populations change slightly over time, with a small pool of tumor cells coexisting with the effector and memory CAR-T cell populations, characterizing a residual disease response; (**c**) 0.2 × 10^6^ CAR-T cell dose is not able to control the tumor, which escapes and reaches the carrying capacity on day 350. The fast decay of effector CAR-T cells prevents the formation of a memory CAR-T cell population. Bottom row: the total CAR-T dose of 2 × 10^6^ cells is fractionated into four equal portions and administered every (**d**) 7 days or (**e**) 14 days; (**f**) the dose is fractionated into three infusions of increasing dose values over 3 days as in [54]. In all cases (**d**)–(**f**), the tumor is eliminated in a few days, followed by a decrease of the effector CAR-T cells. Fractionated infusions lead to the formation of memory CAR-T cells, although the quantity depends on the rest time between doses.

**Figure 5 cancers-13-02941-f005:**
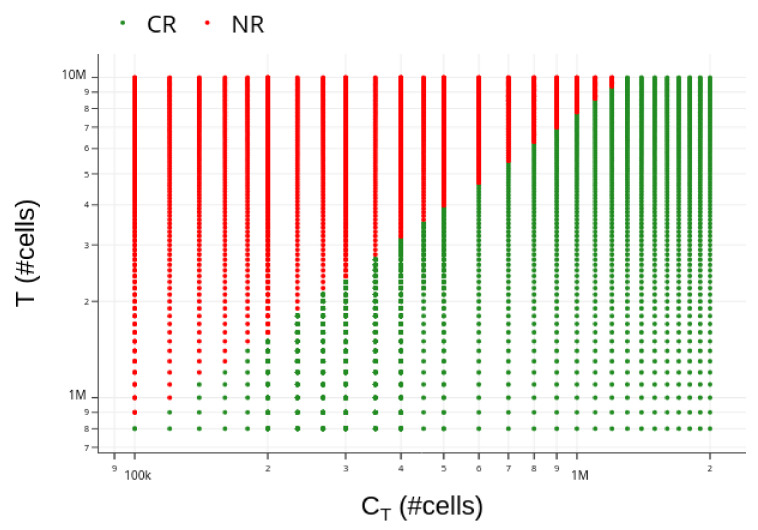
Diagram of occurrence of complete response (CR: $T\left(300\right)\leq 8\times 10^{5}$ cells, green dots) and non-response (NR: $T\left(300\right)\geq 1\times 10^{10}$ cells, red dots) for the HDLM-2 + CAR-T 123 scenario. *T* is the initial tumor burden, and $C_{T}$ is the CAR-T 123 cell dose injected on day 42. The usual ranges of *T* and $C_{T}$ were considered, with the number of tumor cells starting at the detectable limit established in [7] and with the maximum CAR-T cell dose corresponding to the highest value used in [7]. Higher doses ($C_{T}\geq 1.3\times 10^{6}$ cells) are able to eliminate any tumor burden smaller than $10^{7}$ cells. It is worth noting that the CR region decreases with the increase in *T*.

**Figure 6 cancers-13-02941-f006:**
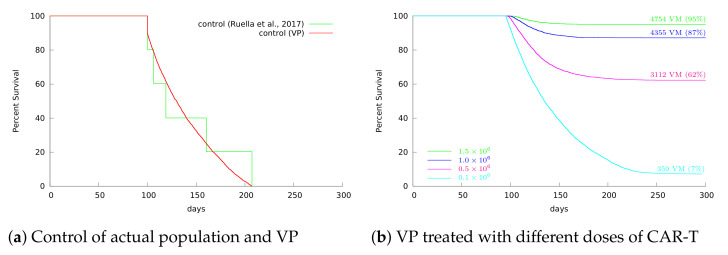
Kaplan–Meier survival curves over 300 days. (**a**) Experimental data from [7] (green) and VP of mice engrafted with HDLM-2 tumor (red). (**b**) CAR-T dose of 1.5 × 10^6^ cells led to 95% overall survival in almost one year, 5% lower than those observed in [7] owing to individual parameter uncertainties. Overall survival decreased significantly with dose reduction. Specifically, the survival rate reached 7% when the dose decreased to 1.0 × 10^5^ CAR-T cells. The number of VM that survived for 300 days for each dosing strategy is also indicated.

**Figure 7 cancers-13-02941-f007:**
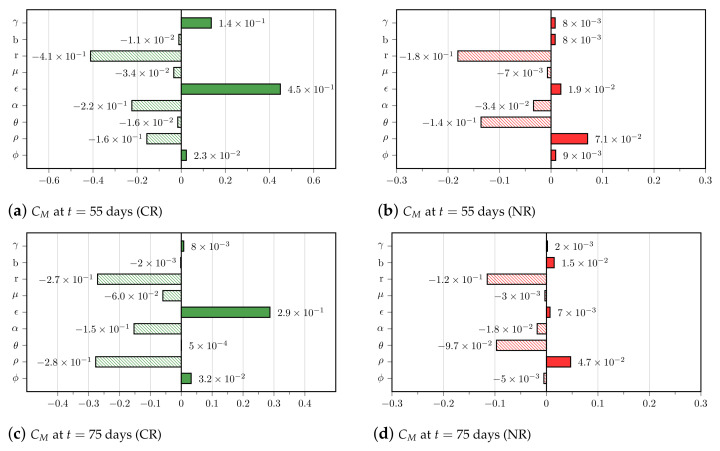
Tornado plots of the Pearson correlation with respect to *C_M_* at *t* = 55 days (top panel) and *t* = 75 days (bottom panel). VP was split into two groups according to the therapy outcomes at 300 days: CR (green) and NR (red). Solid bars indicate a positive effect, while dashed bars indicate a negative one. It is worth noting the important role of the parameter related to memory conversion (ε), which counteracts the intense negative effect of *r* for CR cases. For NR cases, the negative effect of both *r* and θ prevented therapy success. The correlation values at *t* = 75 days are slightly smaller than those at *t* = 55 days, and one may notice a change of order in the rank of the most influential parameters for CR cases.

**Table 1 cancers-13-02941-t001:** Model parameters and the two restrictions imposed on ϕ, ρ, and ϵ.

Parameter	Unit	Meaning
ϕ	day${}^{-1}$	Proliferation rate of effector CAR-T cells
ρ	day${}^{-1}$	Reduction rate of effector CAR-T cells, encompassing the natural death of these cells and their differentiation into memory CAR-T cells
θ	(cell · day)${}^{-1}$	Conversion coefficient of memory CAR-T cells into effector CAR-T cells due to interaction with tumor cells
α	(cell · day)${}^{-1}$	Inhibition/expansion coefficient of effector CAR-T cells due to interaction with tumor cells
ϵ	day${}^{-1}$	Effective conversion rate of effector CAR-T cells into memory CAR-T cells
μ	day${}^{-1}$	Death rate of memory CAR-T cells
*r*	day${}^{-1}$	Maximum growth rate of tumor cells
*b*	cell${}^{-1}$	Inverse of the tumor carrying capacity
γ	(cell · day)${}^{-1}$	Cytotoxic coefficient induced by effector CAR-T cells
**Restriction**	**Meaning**	
ϕ<ρ	Effector CAR-T cells decay to zero in the absence of tumor cells	
ϕ>ρ−ϵ	Healthy donor CAR-T cells proliferate in vivo and differentiate into memory CAR-T cells	

**Table 2 cancers-13-02941-t002:** Model parameter values used in the two immunotherapy scenarios. Additionally, in the RAJI-IDO${}^{{}^{+}}$ scenario, the parameters have the same values as for RAJI-control, except for the α parameter, with $\alpha =1.461699\times 10^{-8}$ (cell·day)${}^{-1}$ for RAJI-IDO${}^{{}^{+}}$ + CAR-T 19 and $\alpha =1.261662\times 10^{-8}$ (cell·day)${}^{-1}$ for RAJI-IDO${}^{{}^{+}}$ + CAR-T 19 + 1-MT. Calibrated parameters are indicated with ${}^{*}$.

Parameter	HDLM-2 + CAR-T 123	RAJI-Control + CAR-T 19
ϕ	0.265 day${}^{-1}$	0.830 day${}^{-1}$
ρ	0.350 day${}^{-1}$	0.8300536 day${}^{-1}$
ϵ	0.150 day${}^{-1}$	1.59795 day${}^{-1}$
θ	$6.0\times 10^{-6}$ (cell · day)${}^{-1}$	$2.3\times 10^{-4}$ (cell · day)${}^{-1}$
α	$4.5\times 10^{-8}$ (cell · day)${}^{-1}$	$1.248506\times 10^{-8}$ (cell · day)${}^{{-1}^{*}}$
μ	$5.0\times 10^{-3}$ day${}^{-1}$	$6.89\times 10^{-7}$ day${}^{-1}$
*r*	$5.650026\times 10^{-2}$ day${}^{{-1}^{*}}$	0.5071721 day${}^{{-1}^{*}}$
*b*	$1.404029\times 10^{-12}$ cell${}^{{-1}^{*}}$	0 cell${}^{-1}$
γ	$3.715843\times 10^{-6}$ (cell · day)${}^{{-1}^{*}}$	$3.365388\times 10^{-8}$ (cell · day)${}^{{-1}^{*}}$

## Data Availability

The data presented in this study are openly available in [7] and [11] at 10.1158/2159-8290.CD-16-0850 and 10.1182/blood-2015-01-621474, respectively.

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
