# Peer review of "CARTmath—A Mathematical Model of CAR-T Immunotherapy in Preclinical Studies of Hematological Cancers"

_cancers, 2021, doi:10.3390/cancers13122941_

Round 1

Reviewer 1 Report

This is a nice study about a mathematical model of CAR T-cell therapy applied to an experimental setting (immunodefficient mice). Because the mice are lacking a functioning immune system, the mathematical model does not need to include competition between the endogenous and engineered T-cells. The paper is written well and easy to read, and I do not have any major concerns. I would only ask that the authors are clear about the applicability of their model and findings to the human setting - they should comment on whether they think that their findings will be applicable beyond the preclinical setting they apply it to and under which conditions.

Author Response

Thank you for your thoughtful reminding. Our model can not be directly applied to human settings since we consider only immunodeficient mouse models (NSG and SCID/beige), which lack adaptive immune systems and have impaired innate immune cells. In this way, our model would be unable to consider a more complex scenario as human patients. For example, healthy B cells are known to play important role in the CAR-T cell expansion in vivo because they also present CD19 antigen (in CAR-T19 scenario), but they are not considered in our model. Macrophages are also important players, especially in cytokine release, which was not considered in our model. There are no CAR-T cells off-target effect on mice, which lack human antigens. Concerning all these issues and being conservative we did not consider our model proper for the human setting. In the revised manuscript, we added a sentence in the discussion section to clarify this topic as followed “Our model does not directly apply to human settings since we do not consider other immune system constituents, especially healthy B cells. By expressing the same tumor antigens, healthy B cells are killed by effector CAR-T cells and seem to play an important role in the in vivo expansion of the CAR-T cells”.

Reviewer 2 Report

In the manuscript “CARTmath - A mathematical model of CAR-T immunotherapy in preclinical models” Luciana R. C. Barros and Colleagues described a developed mathematical model to explore the tumor response to CAR-T cells in immunodeficient mouse models. Particularly, this model encompasses: tumor cells, effector CAR-T cells, and interestingly account also for the conversion of effector cells into memory CAR-T cells. The Authors sustained that the here proposed model is suited to: reduce and optimize the number of in vivo experiments as well as first-rate specific scenarios to be validated in vivo .

The Authors here propose to validate their model via showing how it performs with experimental data from two already published paper (refs 7 and 11): 1) CAR-T 123 cell immunotherapy in Hodgkin’s Lymphoma (slow growing HLDM-2 cell line susceptible to immune rejection) and CAR-T 19 cells immunotherapy on ALL-B (fast growing RAJI cell line not fully eliminated by immune cells) in immunodeficient mouse models.

Albeit the model fits with these two very different scenarios the here presented platform and thus the whole manuscript would greatly benefit from showing in vivo validation data of CARTmath de novo forecasted CAR-T schedule treatments. For example, the dynamic conversion into memory CAR-T cells predicted by the model as well as the in vivo effects of single and fractionated dose treatment. As it is presented right now the model is of limited utility.

As the Authors pointed out the model is far from be applicable to solid tumor in which CAR-T cell activity is also subjected to tumor microenvironment-derived suppressive events, infiltration and spatial distribution of effector cells versus tumor cells. Thus, I advise the title should highlight such restriction.   

Reviewer 3 Report

This manuscript presents a well structured study to simulate CAR-T immunotherapy in mouse models. The methods are adequately described, and the results support the goals of the study. The conclusions follow from the results. 

Author Response

Thank you for the kind revision and comments on our work.

Round 2

Reviewer 2 Report

accept in present form